# Institutional Ownership and Investment Efficiency: Evidence from Iran

**Mohammad Moradi [1], Hassan Yazdifar [2] , Hoda Eskandar [3],\* and Navid Reza Namazi [4]**

[1]  Department of Accounting, University of Tehran, Tehran 14179-35840, Iran; moradimt@ut.ac.ir
[2]  Department of Accounting, Finance and Economics, Bournemouth University, Bournemouth BH12 5BB, UK; hyazdifar@bournemouth.ac.uk
[3]  Department of Accounting, Allameh Tabataba'i University, Tehran 14179-35840, Iran
[4]  Department of Accounting, Shiraz University, Shiraz 71946-84471, Iran; nnamazi@rose.shirazu.ac.ir
**\***  Correspondence: heskandar@ut.ac.ir

**Abstract:** Investment efficiency shows how well a company invests its assets. Although institutional shareholders play undeniable roles in companies, it is not clear whether they are able to monitor managers and make investment decisions or not. This study gives answers to stakeholders, addresses concerns about the effect of the owners on investment efficiency, and aims to add to the literature on emerging markets by investigating the relationship in Iran, a different environment from developed ones. Based on monitoring power, the shareholders are divided into two types: active and passive ones. Investment problems are classified into two types: over- and under-investment problems. The sample consists of 101 firms listed on the Tehran Stock Exchange between 2010 and 2016. Some regression models are used. The results illustrated that institutional owners have a positive effect on investment efficiency and decrease both over- and under-investment problems and so, the efficient monitoring school is approved. Additionally, active ones are positively correlated with investment efficiency and decrease both investment inefficiency problems. Institutional ownership is the cause of investment efficiency, not the reverse. Based on findings, in emerging markets like Iran's market, investors are recommended to give notice to the level of active ownership in firms; ownership structure is a good sign of efficiency.

**Keywords:** active institutional owners; investment inefficiency problems; under-investment; over-investment

## 1. Introduction

Investment efficiency is a benchmark in determining how well a company invests its assets. It is a determinant of the growth and future cash flow of firms (Sun 2014). High investment efficiency indicates that the asset has been used by the company more effectively, which will have a better effect on company performance, so that it can be used as a measure of company performance (Chen et al. 2017). There are several factors involved in company performance. In the findings of Nguyen-Anh et al. (2022), intangible assets are effective in performance. This effect varies in companies with different sizes. Institutional shareholders and corporate governance affect different aspects of the company and its strategies. Therefore, this factor is the source of most changes in the company so that it overshadows all the variables affecting performance.

Prior studies indicated that the problems of asymmetric information and agency have a major impact on investment efficiency (Chen and Yu 2012) due to conflict between the interests of shareholders and managers; also, conflict between majority and minority shareholders leads to a reduction in the efficiency of corporate investment. Many studies showed that ownership structure deals with agency problems through implementing efficient investment decisions and also improves firm performance (Chen and Yu 2012). Institutional ownership is one of the most important parts of a company's structure of ownership.

Institutional ownership refers to the ownership shares in a company held by large financial institutions, pension funds, or endowments. They have an undeniable effect on companies since they possess substantial shareholdings and so, they are supposed to be able to monitor managers. The presence of such owners is likely to change a company's behavior (Velury and Jenkins 2006); nevertheless, their effectiveness in supervising managers has been a critical question. In this regard, there are three hypotheses: (1)—Efficient supervision hypothesis: In this hypothesis, due to risk considerations and high cost of supervision, institutions are more willing to supervise managers and have sufficient motivation to do so. (2)—Conflict of interest hypothesis: This hypothesis states that due to the origin of the behavior of institutional shareholders in the theory of representation, this group will vote for the current management due to conflict of interest. (3)—Strategic alignment hypothesis: According to this hypothesis, institutional shareholders form a kind of strategic alliance with the current management. According to the statements, institutional shareholders influence company behavior, but it is not clear how it does.

This study provides some insights into the monitoring role of institutional shareholders. Some research has been conducted about these issues; nevertheless, this study differs from them in several ways. Firstly, nonetheless, the relationship between institutional ownership, its type, and investment efficiency has not been sufficiently explored and the empirical results obtained so far have shown mixed evidence (even in developed countries). Additionally, few papers addressed such relations in emerging markets. There is a lack of enough support to analyze this relation in the perspective of emerging markets. Emerging economies are not as advanced as EU and other developed countries. Capital markets in such countries have different characteristics in comparison with developed ones (Alawi et al. 2022; Arouri et al. 2013). Emerging economies are in a delicate stage of development in which institutional ownership does not have the same characteristics as ones in developed countries. Difference in institutional owners between Iran and others is likely to result in a different relationship between them and investment efficiency. As a result, due to the sensitive nature of financial markets and institutional ownership in developing economies, this research adds to the current domain of research by offering evidence from emerging regions. So, this study is novel and opens up a new way of studying the institutional shareholders that have not been examined.

This study seeks to answer the questions of whether there is a relationship between institutional investors and investment efficiency. What is the impact of each active and passive institutional investor on investment efficiency? Are the types of institutional investors affecting over-investment and under-investment? This research seeks to answer the questions in a developing country, Iran, a country with different characteristics than developed ones. The sample includes 101 firms listed on Tehran Stock Exchange (excluding financial firms) between 2010 and 2016. Some linear regression models are used and a causality test is examined.

Shortly, the findings showed that institutional ownership is positively correlated with investment efficiency and makes it better. Moreover, all owners are not the same; among them, only active ones with a long-term orientation have a positive effect on investment efficiency and solve both under- and over-investment problems. Additionally, institutional ownership affects investment efficiency, not the reverse. The results will make investors and stakeholders aware of the fact that in emerging markets like Iran's market, institutional owners, particularly those ones who have some representation on the board of directors (active), can decrease investment inefficiency.

The rest of the paper is organized as follows: theoretical foundations are explained in the next section. Then, literature is reviewed. After stating the research method and testing the hypotheses, the findings are presented. Finally, the conclusion is stated and some suggestions are presented.

## 2. Theoretical Foundations of Research

Ownership structure affects companies' investment decisions. Ownership structure has ramifications for managerial decisions, and so influences investment efficiency (Chen et al. 2013). Institutional ownership is one of the most important parts of a company's structure of ownership and so, may affect investment efficiency.

Institutional ownership is the amount of a company's available stock owned by mutual or pension funds, insurance companies, investment firms, private foundations, endowments, or other large entities that manage funds on behalf of others (Bushee 1998). Institutional shareholders are able to access information in a timely manner compared to non-founding shareholders. Founding shareholders have the ability to control the entry and exit of money with no ability to influence the share price, which leads to an increase in the voting rights that they have, but this may be generated by the emergence of agency problems as a result of the omission of legal protection for minority shareholders. They have the ability to monitor the executive management of the company due to their ability to deliver information to shareholders and monitor the organization performance in an efficient manner, which is reflected in the financial performance of the company and leads to increased efficiency. Institutional owners have an important supervisory role in reducing agency costs. The institutional owners appoint the board of directors to serve their interests and are able to control the administration and then improve the current financial performance. Institutional owners have an important role within the board through voting on important decisions that serve the company or refraining from decisions that are harmful to its wealth (Rashed et al. 2018). As a result, such ownership is likely to affect investment decisions.

Although there are some research studies about this effect in developed countries, this study examines this effect in Iran, a country with a different environment in comparison with developed ones.

Iranian institutional owners have features that are interesting and this is the reason why Iran has been selected for this study. They are not the same as in other countries. They are affiliated with state institutions. In contrast to Iran, they are private institutions in developed countries. In the Tehran Stock Exchange (TSE), the number of foreign companies is very scarce compared with others. Such ownership in Iran is less than one percent, in comparison with developed countries, where it is about 50% or more, because accessing the Iran market is too difficult for Western investors (Ghauri 2015; Mehrani et al. 2017).

Today, the number of institutional shareholders has increased in most countries, especially in developed countries. Moreover, in addition, a number of new institutions have entered the market and are the main owners of companies, along with institutional investors who have long been business partners. Pension funds and mutual funds are examples of these groups (Li et al. 2006). Due to having less background, owners of institutions in Iran are less diverse and have fewer experts than developed countries. As a clear example, in Iran, investment funds also have a short history. In recent decades, institutional investors in Iran have grown rapidly as a result of the start of privatization and the creation of the first stock exchange law. The sale of shares of state-owned companies to the public through the stock exchange has begun in recent years with the aim of fulfilling Article 44 of the Iranian constitution. In every initial public offering in the market, institutions are the main buyers. In this regard, the supervision hypothesis proposes that institutional shareholders have a supervisory role in each company.

In developed countries, in addition to the stated content, institutional investors play a major role, and many companies consider joint venture funds and pension funds as their owners. The scattering of ownership structures motivates investors to have effective roles. Therefore, they are likely to help reduce the optional management problem (Li et al. 2006). However, the increasing concentration of corporate ownership, pyramid schemes, and weak legal protections of Iranian owners have discouraged some institutional owners from having sufficient power to control large shareholders. The entanglement of stocks between companies and the separation of voting rights and cash flow rights allow banks or other

non-financial corporations to have a controlling presence and engage in a wide range of actions aimed at harming other institutional investors. As a result, institutional owners can either participate in overseeing management activities aimed at reducing agency disputes or colluding with large shareholders to disqualify minority shareholders (Li et al. 2006; Ghauri 2015).

In Iran, corporate governance mechanisms are weak. In Iran, there is a weak mechanism for corporate governance, so sometimes investors without a representative on the board may have doubts in this unprotected environment. These owners do not have adequate power to have impact on companies' decisions. Conversely, those who are represented on the board have the power to influence corporate decisions (Mashayekhi and Mashayekh 2008). In addition, not all institutions have the same features. Some, like banks, pension funds, and insurance companies, are always represented on boards with the aim of having enough power to oversee managers. Such institutions that are called "active" have control over investors and are long-term-oriented. Institutional shareholders, because they are very large and manage the markets, are not affected by the subsidiaries and companies that have invested, and due to their relative independence, have better oversight and control over managers. They are passionate about affecting companies' decisions as well. Conversely, other natural or legal persons who do not have any representation on the board do not have enough power to control the managers. They are unlikely to challenge managerial decisions. Such owners are "passive" and short-term-oriented (Mehrani et al. 2017; Ghauri 2015).

To sum up, due to the difference between Iran and other countries in terms of institutional ownership, this study is interesting. In addition, along with differences in owner's types, their monitoring roles are likely to be different. Because the nature of the capital market is similar in most countries and corporate governance in companies plays an important role, the findings of this study can be applied to other financial markets in addition to Iran.

## 3. Literature Review

There are various theories about investment efficiency and institutional ownership that are described below.

### 3.1. Investment Efficiency

Investment activities play an important role in firms' operation. In other words, investment efficiency is a signal of the firm's performance. Without any financial market imperfection, all investment opportunities with positive net present value are provided and implemented by companies. In fact, financial market failures are severe, and corporate resource allocation can be inefficient. When the marginal return on investment is equal to the sum of its final cost and the cost of capital adjustment, the optimal level of investment is gained. However, the firm's actual investment always deviates from the optimal state due to friction in capital markets such as external financing costs, conflict of interest between managers and stakeholders, and information asymmetry (Yildiz 2021).

Neo-classical theory states that companies invest until the marginal benefit equals its marginal cost (Abel 1983). Conversely, Keynesian theory states that in order to make an investment, there must be two dimensions of capital security and appropriate growth estimates (Gordon 1992). Moreover, agency theory asserts that companies may deviate from optimal levels of investment and suffer from more or less investment. In perfect financial markets, every project which has positive net present value (NPV) is supposed to be completed. However, prior literature flatly contradicts such an assumption. Market imperfections can result in negative NPV projects being completed (over-investment) and the rejection of positive NPV projects (under-investment). Based on agency theory, both are explained despite the asymmetric information among the shareholders. Jensen (1986) provides a framework in which the role of asymmetric information is considered. In this case, the efficiency of investment is examined through information problems in order to minimize the moral risk and undesirable choice and to maximize the efficiency

of investment. The moral hazard problem arises if there is a mismatch between the interests of the shareholders and there is no oversight of the managers, and therefore it may lead to the management's desire to maximize personal interests by making investments that are not suitable for the owners (Jensen 1986), with subsequent over-investment (Hope and Thomas 2008). Adverse selection arises when managers are better informed and so, they may over-invest if they sell expensive securities and raise extra funds. To prevent this from happening, capital suppliers can offer quota capital or increase its cost. This will lead to the rejection of some profitable projects due to budget constraints (Lambert et al. 2007), which results in low-dimensional investment.

### 3.2. Investment Efficiency and Institutional Ownership

Prior research showed that the higher the inefficient investment, the lower the subsequent company performance (e.g., Titman et al. 2004; Sakaki and Jory 2019). As a result, owners tend to control the managerial investment decisions. Chen et al. (2013) showed that corporate governance and ownership structure have significant effects on corporate investment decisions. Sun (2014) found that ownership structure leads to investment efficiency.

Institutional owners are always being more active in controlling activities in comparison with individual ones. This is because ownership activity is costly and it is difficult for individuals to intervene collectively. However, given the trade-off between the benefits and the costs of active oversight, institutional investors may not have the same incentive to improve corporate governance for the following two reasons. Firstly, institutional investors are heterogeneous at the portfolio level. They vary in type, length of trading horizon, and activity (Bushee 1998; Chen et al. 2007; Schmidt and Fahlenbrach 2017). Secondly, investors' attention is limited. They are not able to monitor all the companies in their portfolio (Kempf et al. 2016); the motivation for institutional oversight therefore depends on the importance of an individual stock in their portfolio. Fich et al. (2015) stated that institutional owners are more motivated to monitor when the target stock is more important than their portfolio.

Two different thought schools are related to the monitoring role of institutional owners. On the one hand, the efficient monitoring school argues that these owners, compared to small individual ones, have more expertise, and so they are able to monitor managers actively (Dau et al. 2020). Their size and information advantage (e.g., research quality, ability to collect and process information) give them strong incentives to control activities (Shleifer and Vishny 1986). Cao et al. (2020) examined the influence of these investors on firm investment efficiency based on non-financial firms listed on Chinese stock exchanges over the period 2009–2014. Their results approved efficient monitoring of managers. Fung and Tsai (2012) investigated the role of institutional investors in improving firm performance through the channel of corporate investment decisions and documented that the interaction effect between institutional ownership and capital expenditures is significantly related to firm performance. On the other hand, according to the school of private interest, larger investments by institutional owners provide an opportunity to access private information. This information can be misused for profit-seeking behavior by institutions that view it as short-term (Koh 2007). Consequently, concentrated ownership in the hands of such owners is supposed to reduce investment efficiency. In addition, they might not monitor investment projects because of factors such as free riders and their relationships with companies' managers. Rashed et al. (2018) showed that institutional ownership, block holder ownership, and outside director ownership have a negative relationship with investment efficiency. Ferreira and Matos (2008) conducted a study to determine whether there is a relationship between institutional ownership and investment efficiency in addition to the examination of the role of institutional ownership around the world. Results indicated that there is a negative relationship between institutional ownership and investment efficiency, leading to an increase in the firm value. Enriques and Romano (2019) documented that institutional owners vote according to their economic interests because of their complexity and information advantage. Nevertheless, mutual arguments argue that the short-sighted

behavior of these owners can lead to their inaction in relation to corporate governance (McConnell and Servaes 1990). Having a dispersed ownership structure can motivate institutional owners to collude with managers to extract private benefits, and can help consolidate current managers (Trąpczyński et al. 2020). Ward et al. (2017) found that American companies with highly motivated regulatory ownership owned less than expected investment levels. Institutional owners, according to the oversight hypothesis, have a higher incentive to oversee the company's performance. This is conducted regardless of over-investment or under-investment. This finding exists in different types of organizations. In addition, we show that motivated regulators reduce over-investment, free cash flow, and under-investment due to managers' job concerns. Companies benefit from firm oversight because our inefficient investment proxies are associated with subsequent stock returns. In general, our results offer an approach regarding the importance of institutional attention in companies and activities.

Chen and Chen (2017) documented a positive relationship between investment efficiency and institutional ownership. They report that the post-purchase investment allocation process is more compatible with creating value for different buyers that have their own governance structures. Moreover, according to the study of Cheng and Zhang (2022), the ability of managers affects company risk and ultimately this has a significant role on company performance. In this regard, the impact of corporate governance is also significant and should be considered. In this regard, the greater the independence of the board of directors and audit committees of companies accepting capital, the lesser the role of corporate governance. In this regard, share ownership by shareholders, managers of various departments, and CEOs is also essential. According to the above discussion, the first hypothesis is:

**Hypothesis 1.** *"There is a significant relationship between institutional ownership and investment efficiency."*

Previous literature showed that owners are not the same. As a prime example, Duggal and Millar (1994) showed that the regulatory role of institutions depends on their type. Different owners pursue different goals and styles, are subject to different legal constraints, and face different competitive pressures depending on their markets. Each person's personality traits, such as risk perception, returns, investment horizons, and the governing role of owners, are influential. As a result, not all organizations are the same, and the role of corporate governance must be considered in each company (Cornett et al. 2007; Bushee 1998).

This study classifies the owners into passive and active ones, based on their ownership power; active persons are represented on the board with the aim of having sufficient power and a desire to control the managers. These investors pay attention to managers and expect to be held accountable according to the principle of accountability. Therefore, their presence increases the efficiency of investment decisions (Cornett et al. 2007). Such owners who have higher shareholdings are more stable than others; they tend to hold shares longer and thus have a more intensive effect on firm investment efficiency (Cao et al. 2020). Owners are prepared to encourage opportunistic management actions in the event of significant abnormal returns. They have a high portfolio turnover and can easily cash in on their investments if the company has low profits and poor performance; institutional shareholders are reluctant to turn their attention there (Maug 1998; Potter 1992). In investing, if only the profit of the project is considered, the choice of option is mistaken. Therefore, the optimality of the choice is not observed.

Baik et al. (2010) stated that large companies by transitional institutions facilitate the opportunistic behavior of managers. Yan and Zhang (2009) showed that passive institutional investors can trade based on noise or incomplete short-term information signals. This affects the activities that managers perform and thus increases the problems of asymmetric information.

Gaspar et al. (2005) documented passive investors exercising poor oversight, allowing managers to pursue mergers and acquisitions.

Based on the results of Parrino et al. (2003), passive owners often do business with the benefit of the doubt. This is likely to be the case in liquid markets such as the United States, where they can maintain liquidity of their assets and evacuate blocks of ownership without lowering stock prices.

Attig et al. (2012) stated that long-term institutional shareholders protect their investments by imposing disciplinary mechanisms on managers and encourage managers to focus on the long-term value of the company. Elyasiani and Jia (2010) showed that active shareholders continuously increase the motivation of institutions to participate in the supervision of the company. Chen et al. (2007) asserted that shareholders of an institution that intends to hold its shares for a longer period of time have more control over their subsidiaries.

Ward et al. (2017) showed that monitoring institutional motivation has a negative correlation with inefficient company investment. In addition, this indirect relationship is strong for different types of institutional owners.

Wong and Yi (2015) found that the institutional ownership of the company as a whole is positively related to the investment efficiency of the company, and this relationship is greater for active investors than for all types of institutional investors.

Trapczyński et al. (2020) stated that institutional shareholders who intend to hold their shares for a longer period of time tend to hire more experienced managers.

Cao et al. (2020) found that only pressure-resistant institutional ownership with long-term perspectives increases firm investment efficiency by alleviating both over-investment and under-investment and the channels through which they improve firm investment efficiency should be of interest to investors, regulators, and academics. According to previous studies, institutional shareholders are expected to influence investment performance due to the oversight theory, which is the controlling role of institutional shareholders in various aspects of the company. This effect is not clear in different types of institutional shareholders. Therefore, different groups of institutional shareholders (active institutional ownership and passive institutional ownership) should be considered in this regard. Accordingly, each group is tested on a separate hypothesis. Given this explanation, the second and third hypotheses are as follows:

**Hypothesis 2.** *"There is a significant relationship between active institutional ownership and investment efficiency."*

**Hypothesis 3.** *"There is a significant relationship between passive institutional ownership and investment efficiency."*

The next hypotheses investigate the inefficient investment directions: under-investment and over-investment.

On the one hand, previous studies document two agency problems which result in under-investment. Firstly, finding a suitable project with positive NPV is time-consuming and takes management effort. If there is no corporate governance system and management oversight is weak, managers may be reluctant to take on too much work pressure and prefer to avoid risk (Bertrand and Mullainathan 2003). Secondly, the returns of new projects are uncertain. As a result, managers may not invest in some projects with positive NPV, aiming to prevent loss. Aghion et al. (2013) documented that institutional owners are likely to decrease managers' concerns. In addition, Hennessy (2004) stated that debt overhang problem leads to under-investment as well. Institutions are able to decrease such a problem by reducing a company's debt borrowing cost.

On the other hand, there are some factors leading to over-investment. Jensen (1986) stated that the empire-building tendency of managers results in over-investment. Blanchard et al. (1994) showed that companies over-invest money from cash windfalls.

According to the results of Harford (1999), companies that have more cash holdings have a strong inclination to make acquisitions with worse subsequent operation performance. However, Richardson (2006) argued that companies which have positive free cash flow are passionate about over-investing their money. Titman et al. (2004) found a negative correlation between over-investment and stock returns. This means that over-investment by managers is not in the interest of shareholders. It is expected that a company with more highly motivated monitoring institutional shareholders will exhibit less over-investment. Ward et al. (2017) illustrated that such shareholders are correlated with both over- and under-investment. This paper identifies institutional ownership as a new factor that reduces a company's inefficient investment in both directions, leading to the following hypotheses:

**Hypothesis 4.** *Institutional ownership reduces the under-investment problem.*

**Hypothesis 5.** *Active institutional ownership reduces the under-investment problem.*

**Hypothesis 6.** *Passive institutional ownership reduces the under-investment problem.*

**Hypothesis 7.** *Institutional ownership reduces the over-investment problem.*

**Hypothesis 8.** *Active institutional ownership reduces the over-investment problem.*

**Hypothesis 9.** *Passive institutional ownership reduces the over-investment problem.*

Besides institutional ownership, other factors also have an effect on investment efficiency. For instance, according to Ward et al. (2017) and Stoughton et al. (2016), the firm size and leverage are correlated with investment efficiency. As a result, these control variables have been introduced to capture their impacts.

## 4. Materials and Methods

All companies listed on TSE (excluding financial firms) are included in the sample. The sample's time period consists of the years between 2010 and 2016. Additionally, the sample includes the firms which have their financial year ending at the Iranian calendar year end. Lastly, the sample includes 101 year firm observations analyzed via Eviews 10.

*4.1. Dependent Variable: Proxy for Investment Efficiency*

Biddle et al. (2009) utilized a model that predicts investment in terms of growth opportunities (measured by sales growth). The model deviations, as reflected in the error term, represent the investment inefficiency.

$$Inv_{I,t} = \beta_0 + \beta_1 \, SG_{i,t-1} + \varepsilon_{i,t} \tag{1}$$

where:

> *Inv*: total company's investment, defined as the net increase in tangible and intangible assets and scaled by lagged total assets.
> *SG*: the rate of change in firm sales.

The residuals reflect investment inefficiency. Positive ones mean that the company is conducting investments at a higher rate than expected according to the sales growth, so it will over-invest. Conversely, negative residuals show that real investment is less than that expected, so it will represent an under-investment scenario. Both scenarios are inefficient investments. Absolute value of the residual is used as a proxy for inefficient investments.

*4.2. Institutional Ownership and Investment Efficiency*

To test the first three hypotheses (H1, H2, and H3), the following models are used, aiming to capture the impact of institutional owners on investment efficiency:

$$InvInEff_{it} = \beta_0 + \beta_1 RINST_{it} + \beta_2 Size_{it} + \beta_3 Lev_{it} + \varepsilon_{it} \qquad (2)$$

$$InvInEff_{it} = \beta_0 + \beta_1 RAC_{it} + \beta_2 RPASS_{it} + \beta_3 Size_{it} + \beta_4 Lev_{it} + \varepsilon_{it} \qquad (3)$$

where:

*InvInEff*: investment inefficiency, which is the absolute values of the residuals from the investment efficiency model (Equation (1));
*RINST*: residual institutional ownership estimated by model (Equation (6)),
*RAC*: residual active institutional ownership estimated by model (Equation (6)),
*RPASS*: residual passive institutional ownership estimated by model (Equation (6)),
*Size*: The natural logarithm of total assets,
*Lev*: long-term debt scaled by total assets,

This paper identifies institutional ownership as a new factor that reduces inefficient investment in both directions. Hence, in order to test other hypotheses (H4 to H9) in terms of over- or under-investment, the following models are used:

$$Over/UnderInv_{it} = \beta_0 + \beta_1 RINST_{it} + \beta_2 Size_{it} + \beta_3 Lev_{it} + \varepsilon_{it} \qquad (4)$$

$$Over/UnderInv_{it} = \beta_0 + \beta_1 RAC_{it} + \beta_2 RPASS_{it} + \beta_3 Size_{it} + \beta_4 Lev_{it} + \varepsilon_{it} \qquad (5)$$

where:

*Over/UnderInv*: the value of the firm's over-investment or under-investment, which is the absolute value of the positive/negative residuals from the investment efficiency model (Equation (1)).

## 5. Results

*5.1. Descriptive Statistics*

Table 1 illustrates descriptive statistics for the main variables.

**Table 1.** Descriptive statistics.

| Firms | Variables | Mean | Minimum | Maximum | Std. Deviation |
|---|---|---|---|---|---|
| Panel A. Total sample firms | InvInEff | 1.67 | 0.38 | 11.45 | 3.52 |
| | INST | 0.47 | 0.00 | 0.97 | 0.33 |
| | AC | 0.30 | 0.00 | 0.97 | 0.28 |
| | PASS | 0.17 | 0.00 | 0.39 | 0.11 |
| | Size | 26.30 | 19.92 | 32.47 | 2.11 |
| | Lev | 0.18 | 0.00 | 0.51 | 0.17 |
| Panel B. | **Variables** | **Mean** | **Minimum** | **Maximum** | **Std. Deviation** |
| Under-investment firms | UnderInv | 0.32 | 0.38 | 1.45 | 0.52 |
| | INST | 0.22 | 0.00 | 0.76 | 0.21 |
| | AC | 0.13 | 0.00 | 0.67 | 0.22 |
| | PASS | 0.09 | 0.00 | 0.47 | 0.16 |
| | Size | 19.20 | 19.92 | 32.47 | 2.12 |
| | Lev | 0.16 | 0.00 | 0.32 | 0.09 |
| Over-investment firms | OverInv | 1.35 | 1.08 | 11.45 | 3.01 |
| | INST | 0.26 | 0.00 | 0.97 | 0.32 |
| | AC | 0.17 | 0.00 | 0.97 | 0.28 |
| | PASS | 0.09 | 0.00 | 0.43 | 0.10 |
| | Size | 33.30 | 22.94 | 30.50 | 2.02 |
| | Lev | 0.19 | 0.05 | 0.51 | 0.19 |

The mean institutional ownership as a whole, active, and passive is 47%, 30%, and 17%, respectively. It means that institutions, on average, hold approximately 47% of the total shares outstanding of the sample firms. Active owners own 30% and passive owners own much less (approximately 16%). The average of investment inefficiency, under-investment, and over-investment is 1.67, 0.32, and 1.35, respectively. Mean long-term debt (18%) shows that Iranian companies do not heavily rely on debt, and so their default risk is supposed to be low. In Iran, firms have few long-term debts because of the forbiddance of bonds. According to the table, LEV of over-investment firms is more than under-investment firms. Therefore, firms with over-investment are more financed by long-term debts. The firms' average size is approximately 26.30. The over-investment firms are larger than under-investment ones.

*5.2. Endogeneity*

The characteristics of the company are related to institutional ownership so that institutional ownership interacts with the endogenous characteristics of the company. This feature can create a two-way mechanism and interact simultaneously. To the extent that these economic factors that determine institutional ownership also explain investment efficiency, they can introduce a false relationship between investment efficiency and institutional ownership. To alleviate this concern, all analyses are performed using a measure of residual ownership, defined as the residual property regression on the economic factors that determine it.

Consistent with Ramalingegowda and Yu (2012), ownership fluctuations are attributed to four sources, and multiple proxies are used to capture them. First, institutions as trustees prefer companies that the court finds to be a prudent investment. This "caution" incentive is characterized by company age, dividends, membership in the top 50 TSEs, and stock price fluctuations. Second, because firms are more inclined to invest larger amounts, they prefer companies with high liquidity and low transaction costs. These factors are calculated by company size, stock price, and stock turnover. Third, institutions are interested in investing in companies based on historical return patterns. This priority is determined by size, book-to-market ratio, and momentum. Lastly, the influence of growth options and information asymmetry is captured by Tobin's Q and bid-ask spreads.

$$Own_t = \beta_0 + \beta_1 MB_{t-1} + \beta_2 ME_t + \beta_3 V_{t-2,t} + \beta_4 T_{-3} + \beta_5 P_t + \beta_6 TSE_{50} + \beta_7 M_{-3,0} + \beta_8 M_{-12,-3} + \beta_9 Y_t + \beta_{10} D_t + \beta_{11} S_{t-1} + \beta_{12} TOBINSQ_{t-1} + \varepsilon_t \qquad (6)$$

where:

> *Own*: percentage of common shares held by institutional owners (INST), active institutional owners (AC: institutions with representation on board of directors) or passive institutional owners (PASS: institutions without representation on board of directors) at the end of the year $t$,
> *MB*: market-to-book ratio,
> *ME*: market value of equity,
> $V_{t-2,t}$: the variance of monthly returns over the previous two years (from year $t-2$ to $t$),
> $T_{-3}$: monthly volume divided by shares outstanding, measured three months prior to the end of year $t$,
> *P*: share price,
> $TSE_{50}$: dummy variable, equal to 1 if the company is included in the TSE top 50 index at the end of year $t$, and 0 otherwise,
> $M_{-3,0}$: company's gross return for the three months prior to the end of year $t$,
> $M_{-12,-3}$: company's gross return for the nine months ending three months prior to the end of year $t$,
> *Y*: the number of years a company is listed in TSE at the end of year $t$,
> *D*: dividends,
> *S*: average of daily bid-ask spread, computed as $(ask - bid)/[(ask + bid)/2]$,

*TOBINSQ*: market value of assets divided by book value of assets, where the market value of assets is computed as book value of liabilities plus market value of equity less the sum of book value of equity and balance sheet deferred taxes.

$\varepsilon$: residual term.

First, ownership (i.e., INST, AC, and PASS) is regressed on a wide range of company features that explain it. Afterwards, the regression residuals are extracted as residual ownership measurements (RINST, RAC, and RPASS). In other words, residuals of ownership are used in all of the analyses instead of raw ownership.

Table 2 illustrates that all types of institutions have a strong inclination to invest more in companies with lower volatility. Nevertheless, there are also significant differences across different types of institutions. For instance, active institutions prefer younger companies with larger spread, while passive institutions invest more in older ones with smaller spread. Additionally, while there is a positive relationship between active ownership and market value, there is not a significant relationship between passive ownership and MV. A positive correlation exists between share turnover and passive ownership, even though there is no relationship for active ownership.

**Table 2.** Endogeneity results (Equation (6)).

| Variable | INST | AC | PASS |
|---|---|---|---|
| (Constant) | 0.43 (8.46) | 0.35 (7.11) | 0.08 (5.09) |
| $MB$ | 0.0 (0.55) | 0.00 (1.44) | −0.00 (−1.87) * |
| $MV$ | 0.05 (5.23) *** | 0.06 (5.49) *** | −0.00 (−1.03) |
| $V$ | −0.00 (−3.95) *** | −0.00 (−3.17) *** | −0.00 (−2.19) ** |
| $T$ | −0.00 (−0.69) | −0.00 (−1.15) | 0.00 (1.66) * |
| $P$ | 0.00 (2.14) ** | 0.00 (1.02) | 0.00 (2.78) ** |
| $TSE_{50}$ | 0.01 (0.52) | −0.01 (−0.35) | 0.04 (2.77) *** |
| $M_{-3,0}$ | −0.00 (−1.11) | −0.00 (−0.92) | −0.00 (−0.55) |
| $M_{-12,-3}$ | −0.00 (−0.11) | −0.00 (−0.33) | 0.00 (0.61) |
| $Y$ | −0.00 (−2.50) ** | −0.01 (−4.76) *** | 0.00 (6.01) *** |
| $D$ | 0.00 (0.67) | 0.00 (0.51) | 0.00 (1.03) |
| $S$ | 3.86 (2.31) ** | 4.79 (3.11) *** | −1.09 (−1.89) * |
| $TOBINSQ$ | 0.03 (1.58) | 0.03 (1.78) * | −0.00 (−0.09) |

*T-statistics* are shown in parentheses. ***, **, * denote significance at 0.001, 0.05, and 0.10 levels, respectively, based on *t*-tests (two-tailed).

## 5.3. Correlation

Table 3 illustrates the Pearson's correlations among explanatory variables. The firm size is positively correlated with LEV.

**Table 3.** Correlation matrix.

| Variable | (1) | (2) | (3) | (4) | (5) | (6) |
|---|---|---|---|---|---|---|
| *RINST* (1) | 1 | 0.97 (0.00) ** | 0.14 (0.00) * | −0.01 (0.84) | −0.05 (0.26) | 0.01 (0.90) |
| *RAC* (2) | | 1 | 0.09 (0.06) | −0.01 (0.84) | −0.03 (0.54) | 0.02 (0.73) |
| *RPASS* (3) | | | 1 | 0.00 (0.99) | 0.06 (0.16) | 0.01 (0.89) |
| *InvEff* (4) | | | | 1 | 0.43 (0.51) | 0.49 (0.30) |
| *SIZE* (5) | | | | | 1 | 0.49 (0.00) ** |
| *LEV* (6) | | | | | | 1 |

***, **, * denote significance at 0.001, 0.05 and 0.10 levels, respectively, based on *t*-tests (two-tailed).

### 5.4. Findings

Table 4 shows the results of Breusch–Pagan (BP) tests and the Jarque–Bera (J-B) test. It illustrates that the variances of the errors in the regression models are constant. The Jarque–Bera (J-B) shows that the sample data have skewness and kurtosis matching a normal distribution. Given that *p*-value of the JB statistics is more than 0.05, the hypotheses based on normality of the dependent variables' distribution at 95% confidence level are approved and this means that the dependent variables have normal distribution.

**Table 4.** Breusch–Pagan and Jarque–Bera test of the dependent variables.

| Test | InvEff | UnderInv | OverInv |
|---|---|---|---|
| P-V(BP) | 0.282 | 0.200 | 0.295 |
| P-V (JB) | 0.532 | 0.400 | 0.293 |

The Tables 5 and 6 represent the results of the regression models. Durbin Watson (D.W) statistics indicate that there is no autocorrelation in the samples. Table 5 shows the results for RINST, and Table 6 shows the results for RAC and RPASS.

**Table 5.** Investment efficiency and institutional ownership.

| Variables | Total Firms (Equation (2)) | Under-Investment Firms (Equation (4)) | Over-Investment Firms (Equation (4)) |
|---|---|---|---|
| RINST (β1) | −2.04 (−3.64) *** | −0.27 (−2.84) ** | −0.12 (−3.70) *** |
| Size | −0.10 (−2.19) ** | −0.06 (−1.33) | −0.00 (−0.67) * |
| LEV | 0.10 (0.22) | 0.23 (0.57) | 0.41 (1.41) |
| $R^2$ | 41% | 11% | 19% |
| P−V (F) | 0.000 | 0.000 | 0.000 |
| D.W | 2.05 | 1.70 | 1.82 |

*T-statistics* are shown in parentheses. ***, **, * denote significance at 0.001, 0.05, and 0.10 levels, respectively, based on *t*-tests (two-tailed).

**Table 6.** Investment efficiency and institutional ownership types.

| Variables | Total Firms (Equation (3)) | Under-Investment Firms (Equation (5)) | Over-Investment Firms (Equation (5)) |
|---|---|---|---|
| RAC ($\beta$1) | −1.94 (−2.88) ** | −0.09 (−2.90) ** | −0.06 (−2.69) ** |
| RPASS ($\beta$2) | −1.54 (−1.18) | −0.09 (−1.50) | −0.06 (−1.40) |
| Size | −0.01 (−2.38) ** | 0.01 (1.36) | −0.01 (−1.04) * |
| LEV | −0.76 (−0.43) | 0.26 (1.60) | −0.31 (−1.23) |
| $R^2$ | 32% | 17% | 15% |
| P−V (F) | 0.000 | 0.000 | 0.000 |
| D.W | 2.02 | 1.80 | 1.77 |

*T-statistics* are shown in parentheses. ***, **, * denote significance at 0.001, 0.05, and 0.10 levels, respectively, based on *t*-tests (two-tailed).

The estimated coefficient of RINST shows that higher institutional ownership is related to higher InvInEff (coeff. = −2.04, $t$ = −3.64). This means that InvInEff is decreased by approximately 2.04 for each percentage point increase in shares held by institutions. In other words, institutional shareholders decrease InvIneff and so increase investment efficiency. Table 5 shows that active ownership has a negative relationship with InvInEff (coeff. = −1.94, $t$ = −2.88), which indicates that investment efficiency is increased by approximately 1.94 for each percentage point increase in active ownership. However, there is not a significant relationship between passive ownership and InvInEff (coeff. = −1.54, $t$ = −1.18). Moreover, the large companies have higher investment efficiency. According to Table 5, institutional ownership decreases both under- and over-investment problems.

Based on Table 6, while active ownership decreases both problems of investment inefficiency, passive ownership is not correlated with the two types of investment inefficiency problems.

The achieved results show a positive correlation between institutional ownership and investment efficiency. This finding is consistent with monitoring institutions demanding high investment efficiency.

*5.5. Causality Tests*

While investment efficiency has been assumed as an outcome of ownership, ownership can be the consequence of investment efficiency as well. Companies that have higher investment efficiency can attract investment by institutions as well. This "reverse causality" explanation is plausible because institutional owners might prefer companies with higher investment efficiency to decrease their own monitoring costs. In addition, investment efficiency and institutional oversight of companies can be achieved simultaneously. It can be concluded that this monitoring can improve efficiency and be accelerated by the characteristics of the company at the same time. Similar to Ramalingegowda and Yu (2012), tests are performed to determine the evidence to determine the cause-and-effect relationship between these two variables. Monitoring institutions' current and lead residual ownership (i.e., RINSTt, RINSTt+1, RACt, RACt+1, RPASSt, RPASSt+1,) are added to the third and fourth models (Equations (3) and (4)), aiming to examine how investment efficiency is related to the remaining, current and leading ownership by regulatory bodies.

Tables 7 and 8 report the results from estimating causality equations.

**Table 7.** Causality tests of institutional ownership.

| InvInEff | RINST | | |
|---|---|---|---|
| | Lagged (t − 1) | Current (t) | Lead (t + 1) |
| InvInEff | −1.94 (−2.94) *** | 1.54 (1.61) | 2.31 (1.77) |
| OverInv | −1.34 (−3.14) *** | 1.13 (1.44) | 0.32 (1.21) |
| UnderInv | −2.10 (−2.87) *** | 1.46 (1.00) | 1.42 (1.09) |

\*\*\*, \*\*, \* denote significance at 0.001, 0.05, and 0.10 levels, respectively, based on *t*-tests (two-tailed).

**Table 8.** Causality tests of institutional ownership types.

| InvInEff | RAC | | | RPASS | | |
|---|---|---|---|---|---|---|
| | Lagged (t − 1) | Current (t) | Lead (t + 1) | Lagged (t − 1) | Current (t) | Lead (t + 1) |
| InvInEff | −2.19 (−3.08) *** | −0.24 (−0.05) | 2.13 (0.98) | 4.67 (1.36) | −1.62 (−0.85) | −0.85 (−0.57) |
| OverInv | −1.98 (−3.01) *** | −0.18 (−0.22) | 1.08 (0.87) | 4.98 (1.05) | −0.74 (−0.90) | −0.88 (−0.59) |
| UnderInv | −2.10 (−3.20) *** | −0.21 (−0.05) | 1.27 (0.85) | 5.70 (1.15) | −0.81 (−0.79) | −1.05 (−0.73) |

*T-statistics* are shown in parentheses. \*\*\*, \*\*, \* denote significance at 0.001, 0.05, and 0.10 levels, respectively, based on *t*-tests (two-tailed).

A positive correlation was found between investment efficiency and lagged residual ownership. Moreover, no significant relationships were found between investment efficiency and both lead and current residuals ownership. However, the results of active ownership are not significantly different from total ownership, and inactive residues are not correlated with investment efficiency. These findings indicate that changes in investment efficiency do not increase institutional ownership. To sum up, there is a causal flow from changes in institutional ownership to changes in investment efficiency.

## 6. Conclusions

Investment decisions are one of the most challenging decisions for managers. The efficiency of such decisions shows that a firm allocates its resources well. According to prior literature in developed countries, institutional owners are correlated with investment decisions because of their monitoring of managers. However, all such owners are not the same. Some of them monitor managers actively because of their long-term orientation. However, others are passive and short-term-oriented. Despite the active ones, passive ones have no sufficient motivation for monitoring managers' decisions. This study seeks to investigate the effect of institutional owners' types on investment efficiency in Iran, a country with unique ownership features, in comparison with developed countries. Ownership structure (including institutional ownership) in emerging markets (like Iran) differs from developed ones. These differences are likely to make the relationship between institutional owners and investment efficiency different. This is the reason why this study is interesting.

In order to examine this issue, investment inefficiency is classified into two problems: under-investment and over-investment problems and their relation with institutional ownership are investigated among Iranian firms listed in TSE between 2010 and 2016.

The results provide some evidence that institutional ownership is positively correlated with investment efficiency and decreases both under- and over-investment problems. In other words, companies owned by high institutional investors deviate less from predicted investment levels. Higher institutional ownership is related to both over- and under-investment. This result is similar to the results of Sun (2014); Ward et al. (2017);

Chen and Chen (2017). Moreover, since all institutional owners are not the same, according to the features of Iranian institutional owners, this study classifies them into active (with at least one representative on the board of directors) and passive groups (without any representative on the board of directors), and then their relationship with investment efficiency is tested again. Findings provide mounting evidence that active ownership has a positive ramification on investment efficiency. Hence, it can solve both investment inefficiency problems. This means that such long-term-oriented institutional investors control corporate investment decisions and motivate higher investment efficiency. These results prove that these powerful investors in Iran have a strong tendency to monitor managers and affect the managers' decisions. In spite of this, passive ones are not related to investment efficiency, provided that such owners are unlikely to have enough power to play any roles in managers' decisions. This result approves the result of Wong and Yi (2015).

In addition, causality tests of variables show that the ownership of the institution that oversees the organizations it owns not only has no adverse effect, but also leads to greater investment efficiency.

Generally, these findings are consistent with monitoring institutions demanding investment efficiency. Given that active institutional owners contribute to investment efficiency, the companies with high levels of such owners are highly likely to have more investment efficiency. This is the reason why investors are recommended to notice the level of active ownership in firms; ownership structure is a good sign of investment efficiency. In other words, investors are supposed to put particular emphasis on taking into account the level of such ownership, aiming to assess the efficiency of firms' investment decisions.

Finally, there are some issues requiring further research and investigation. As mentioned earlier, however, ownership structure in developing countries differs from developed ones; the relationship between institutional ownership types and investment efficiency has not been studied sufficiently in developing countries. Therefore, this is a possible new direction for future work. Moreover, future research should cover longer time periods (in comparison to the 7-year period of this study), aiming to explore long-term relationships between the variables.

**Author Contributions:** Investigation, M.M. and H.E.; methodology, H.Y.; data collection, H.E.; formal analysis, H.E. and M.M.; supervision, H.Y. and M.M. Project administration, H.E. and N.R.N.; writing—original draft, H.E.; writing—review and editing, N.R.N. All authors have read and agreed to the published version of the manuscript.

**Funding:** This research received no external funding.

**Institutional Review Board Statement:** Not applicable.

**Informed Consent Statement:** Not applicable.

**Data Availability Statement:** Data for this study was accessed from Tehran Stock Exchange site (TSETMC).

**Conflicts of Interest:** The authors declare no conflict of interest.

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
