# Peer review of "Institutional Ownership and Investment Efficiency: Evidence from Iran"

_jrfm, doi:10.3390/jrfm15070290_

Round 1
Reviewer 1 Report
This is an interesting paper and I enjoyed reading it. However, there are essential weaknesses that need to be addressed.
0) Abstract: Authors should state their contribution in terms of issue problems solved or ameliorated, theory or policy dilemmas resolved, or the like. Abstract should offer at least one example of a theoretical or managerial implication that authors concluded after their work.
1) The introductory/opening section should communicate a little clearer the literature gaps, as well as the study's aims & objectives in order to facilitate the flow of the study.
2) Overall there are good arguments and well researched points made in this paper, but I feel that author needs to take ´Theoretical development and research hypothesis´ (page 2), and ´Conclusion and discussion´ (page 10) to a further level. The paper is interesting, but there is a lack of development in the theoretical argument and its link with the hypotheses.
It is important to read and cite (where appropriate) current literature, providing a substantial number of citations to support your work. It is also important to read (and, if relevant, cite) papers that have already been published in the Journal of Risk and Financial ManagementThis will help to show the consistency of your research with the debate taking place in the journal.
The author(s) need to invest more effort in developing the linkage between recent theory and hypotheses.
Additional references to recent & relevant empirical studies could increase the quality of the research paper and provide a much clearer message to the reader - these may help you building your discussion which needs to be extended. Add the following to your reference list:
Aibar-Guzmán, B., García-Sánchez, I., Aibar-Guzmán, C., & Hussain, N. (2022). Sustainable product innovation in agri-food industry: Do ownership structure and capital structure matter? Journal of Innovation & Knowledge, 7(1), 100160. https://https://doi.org/10.1016/j.jik.2021.100160
Cheng, L., & Zhang, Y. (2022). Do able managers take more risks? Journal of Innovation & Knowledge, 7(2), 100182. https://https://doi.org/10.1016/j.jik.2022.100182
Giaretta, E., & Chesini, G. (2021). The determinants of debt financing: The case of fintech start-ups. Journal of Innovation & Knowledge. 6(4), 268-279. DOI: 10.1016/j.jik.2021.10.001
Hao, Z., Zhang, X., & Wei, J. (2022). Research on the effect of enterprise financial flexibility on sustainable innovation. Journal of Innovation & Knowledge, 7(2), 100184. https://https://doi.org/10.1016/j.jik.2022.100184
Ortigueira-Sánchez, L. C., Welsh, D. H., & Stein, W. C. (2022). Innovation Drivers for Export Performance. Sustainable Technology and Entrepreneurship, 1(2), 100013. https://doi.org/10.1016/j.stae.2022.100013
Climent, R. C., & Haftor, D. M. (2021). Value creation through the evolution of business model themes. Journal of Business Research, 122, 353-361. https://doi.org/10.1016/j.jbusres.2020.09.007
Some of the statements you make are entirely obvious and should be supported in the text by these specific references.
2) The question could be asked of whether this study is representative of other sectors in your country or in the world. Please explain this potential applicability to a general context.
3) The statistical treatment is acceptable.
4) At the end of the ´Conclusion´ section, the author should include clear statements as to where research should now go – what are the issues requiring further research and investigation? The author has to suggest challenges and possible new directions for future work. Perhaps: if the results obtained are only studied in the short term, which is then an important bias in analysing the influence further than three years in time and in their influence in the future.
4) Concluding remarks – authors must elaborate more on what is their contribution to the literature as well as on opportunities for future research. Questions that need to be answered: Why your study is important? and how it extend so existing knowledge on the issue/topic? Conclusions need to be written in a clear and coherent manner and draw the main lessons from the paper. I suggest you to concentrate on the description of the implications of the work, the main findings and its potential replicability - empirical investigation. Furthermore, limitations of the study need to be outlined to a greater extent, and so are any potential connections between your study and specific aspects of the Journal's scope.
5) Carefully check the references, so as to make sure they are all complete and follow the Guidelines to Authors.
6) Finally, when you submit the corrected version, please do check thoroughly, in order to avoid grammar, syntax or structure/presentation flaws. Make sure you retain a formal/academic-specific style of presenting your work throughout the text - (if necessary) please seek for professional English proofreading services or ask a native English-speaking colleague of yours in order to refine and improve the English in your paper.
6) The paper needs to be revised by an English native speaker. Some expressions need to be revised and given a fresh approach by an experienced native proofreader.
Thank you for the opportunity to read the paper.
Author Response
1.Abstract: some contribution and implications have been stated in abstract:
This study answers to stakeholders, concern about the effect of the owners on investment efficiency and aims to add to the literature in emerging markets by investigating the relationship in Iran, a different environment from developed ones. Based on findings, in emerging markets like Iran’s market, investors are recommended noticing to the level of active owners in firms, ownership structure as a good sign of efficiency.
2.The introduction section communicates the literature gaps, as well as the study's aims & objectives:
Some research has been conducted about these issues; nevertheless, this study differs from them in several ways. Firstly, nonetheless, the relationship between institutional ownership, its type and investment efficiency has not been sufficiently explored and the empirical results obtained so far have shown mixed evidence (even in developed countries). Additionally, few papers addressed such relation in emerging markets. There is a lack of enough support to analyze this relation in the perspective of emerging markets. Emerging economies are not as advanced as EU and other developed countries. Capital market in such countries have different characteristics in comparison with developed ones (Alawi et al. 2022 and Arouri et al. 2013). Emerging economies are in a delicate stage of development in which institutional ownership does not have the same characteristics as ones in developed countries. Difference in institutional owners between Iran and others is likely to result in different relationship between them and investment efficiency. As a result, due to the sensitive nature of financial markets and institutional ownership in developing economies, this research adds to the current domain of research by offering evidence from emerging regions. So, this study is novel and opens a new way of studying the institutional shareholders that have not been examine.
This study seeks to answer the questions of whether that is there a relationship between institutional investors and investment efficiency? What is the impact of each active and passive institutional investor on investment efficiency? Are the types of institutional investors affecting over-investment and under-investment? This research looking for to answer the questions in a developing country, Iran, a country with different characteristics with developed ones.
3.Theoretical foundation was being improved: please refer to the section.
4.Conclusion has been improved to a further level: refer to conclusion section.
5.Some new and relevant reference have been cited such as Alawi, et al. (2022), Arouri, et al. (2013), Cao, et al. (2020), Chen et al. (2017, 2012), Cheng, Zhang, (2022), Ferreira, Matos (2008), Fung, Tsai (2012), Nguyen-Anh, et L. (2022) and Rashed, et al. (2018).
6.At the end of the ´Conclusion´ section, some clear statements have been included to where research should now go – what are the issues requiring further research and investigation? In addition, some possible new directions for future work have been stated:
there are some issues requiring further research and investigation. As mentioned earlier, however ownership structure in developing countries differs from developed ones, the relationship between institutional ownership types and investment efficiency has not been studied sufficiently in developing countries. Therefore, it is a possible new direction for future work. Moreover, future research should cover longer time period (in comparison 7-year period of this study), aiming to explore long term relationship between the variables.
7) Concluding remarks Why your study is important? and how it extend so existing knowledge on the issue/topic? Conclusions was being written in a clear and coherent manner and draw the main lessons from the paper: refer to the conclusion section.
8) All references were being checked and all of them are all complete and follow the Guidelines to Authors.
9) we checked the manuscript thoroughly, in order to avoid grammar, syntax or structure/presentation flaws. Additionally, it was revised by an English native speaker.

Reviewer 2 Report
- The general format of the article don’t address the rules of the journal (eg. don’t appear the word Abstract, don’t appear the line numbers etc.). Should be aligned with the requirements of the journal template
- Usually keywords don't take (over) sequences from the title - I recommend replacing them so that they can reflect the ideas in the article and not just be redundant
- It would be interesting if the study would present some aspects related to the practical application of the study and its results (where could be applied, how could be applied and so).
1. Introduction:
- The section of introduction should include (even briefly ): the context of the study, the gap in literature that the present paper intends to cover, which is the used methodology, which are the main results presented in short, which is the originality of this paper, the main implication policy of these results and a short plan of the rest of the paper. Some of them are missing. I recommend to fill it accordingly
2. Theoretical foundations of research – Ok
3. Literature Review
3.1.Investment Efficiency – ok
3.2.Investment Efficiency and Institutional Ownership
H2 - Even was referred in H1, this hypothesis (H2) is not properly developed. I recommend to fill it.
4. Methodology
4.2. Institutional Ownership and Investment Efficiency - refers to H1 - H3, but as mentioned earlier, H2 is not addressed at all. Consequently, how is the study / methodology based on an unapproached (H2) hypothesis?
5. Conclusion
- The sentences in the Conclusions chapter "repeat" the sentences in the body of the document without referring to concrete elements related to the topic - it would be interesting to introduce a chapter on concrete proposals before the conclusions chapter
- The study is interesting as a way of construction, but has a problem related to the particularity of a market in which the institutional system classifies companies (both in terms of their field of activity and their importance / stock position) and the whole system determines Institutional Ownership and implicitly, can dictate the result of Investment Efficiency. Specifically, all data may be alterated / corrupted (due to specific features and the lack of a strong competitive market and independent control).
- It would be interesting if the study would present some ideas related to the reporting of a possible total lack of institutional investment and the general reaction of the market and companies related to this aspect.
6. References:
The references don’t address the rules of the journal. Should be aligned with the requirements of the journal template
Author Response
General observations:
- The general format of the article addresses the rules of the journal.
- some aspects related to the practical application of the study and its results have been presented in both abstract and introduction:
This study answers to stakeholders, concern about the effect of the owners on investment efficiency and aims to add to the literature in emerging markets by investigating the relationship in Iran, a different environment from developed ones.
The findings will make aware investors and stakeholders of this fact that in emerging markets like Iran’s market, institutional owners, particularly those ones who have some representations in the board of directors (active), can decrease investment inefficiency. Therefore, in emerging markets like Iran’s market, investors are recommended noticing to the level of active owners in firms, ownership structure as a good sign of efficiency.
- Keywords have been stated align with the title.
- Introduction has been addressed the gap in literature that the present paper intends to cover:
Some research has been conducted about these issues; nevertheless, this study differs from them in several ways. Firstly, nonetheless, the relationship between institutional ownership, its type and investment efficiency has not been sufficiently explored and the empirical results obtained so far have shown mixed evidence (even in developed countries). Additionally, few papers addressed such relation in emerging markets. There is a lack of enough support to analyze this relation in the perspective of emerging markets. Emerging economies are not as advanced as EU and other developed countries. Capital market in such countries have different characteristics in comparison with developed ones (Alawi et al. 2022 and Arouri et al. 2013). Emerging economies are in a delicate stage of development in which institutional ownership does not have the same characteristics as ones in developed countries. Difference in institutional owners between Iran and others is likely to result in different relationship between them and investment efficiency. As a result, due to the sensitive nature of financial markets and institutional ownership in developing economies, this research adds to the current domain of research by offering evidence from emerging regions. So, this study is novel and opens a new way of studying the institutional shareholders that have not been examine.
- The methodology, the main results and the main implication has been included shortly in the introduction:
The sample includes 101 firms listed on Tehran Stock Exchange (excluding financial firms) between 2010 to 2016. Some linear regression models are used and a causality test are examined.
Shortly, the findings showed that institutional ownership positively correlated with investment efficiency and makes it better. Moreover, all owners are not the same, among them, only active ones whose long-term orientation, have positive effect on investment efficiency and solve both under and over investment problems. Additionally, institutional ownership affect investment efficiency not reverse. The results will make aware investors and stakeholders of this fact that in emerging markets like Iran’s market, institutional owners, particularly those ones who have some representations in the board of directors (active), can decrease investment inefficiency.
- Conclusion was being rewritten: please refer to the conclusion section.
- References address the rules of the journal.

Round 2
Reviewer 1 Report
Thanks for the opportunity to review the paper
Author Response
Dear reviewer
We appreciate you because of your valuable comments. Some parts of results and conclusion have been rewritten and improved.
Reviewer 2 Report
- Like I said in RV1, the general format of the article don’t address all the rules of the journal (eg. don’t appear the word Abstract, whicn must be “A single paragraph of about 200 words maximum.”) - should be aligned with the requirements of the journal’s template
- Like I said in RV1, usually keywords don't take (over) sequences from the title - I recommend replacing them so that they can reflect the ideas in the article and not just be redundant. Besides this, according with the journal template must “List three to ten pertinent keywords specific to the article” (not just two redundant keywords)
1. Introduction – quite ok
2. Theoretical foundations of research – ok
3. Literature Review - ok
4. Material and Methods - ok
5. Results - quite ok (the situation is really particular / specific and therefore the results are particular - it relates to particular situation/ s)
6. Conclusion - quite ok (same previous explication). Besides this, like I said in RV1, some sentences in the Conclusions chapter "repeat" some sentences from the introduction or the body of the document without new elements – should be reformulated in the way to introduce new data / information
7. References: Not all references address the rules of the journal - should be aligned with the requirements of the journal’s template
Author Response
Some parts of results and conclusion have been rewritten and improved.
Manuscript addresses all the rules of the journal.
According to the journal requirements, abstract has been modified.
Keywords have been revised.
References have been checked. They are aligned with the journal template.